# Antimicrobial Effects of Potential Probiotics of *Bacillus* spp. Isolated from Human Microbiota: In Vitro and In Silico Methods

**DOI:** 10.3390/microorganisms9081615

**Published:** 2021-07-29

**Authors:** Alfonso Torres-Sánchez, Jesús Pardo-Cacho, Ana López-Moreno, Ángel Ruiz-Moreno, Klara Cerk, Margarita Aguilera

**Affiliations:** 1Department of Microbiology, Faculty of Pharmacy, University of Granada, Campus of Cartuja, 18071 Granada, Spain; alfons_ats@hotmail.com (A.T.-S.); jesuspar99@correo.ugr.es (J.P.-C.); angel_trm_@hotmail.com (Á.R.-M.); klara.cerk@gmail.com (K.C.); 2Institute of Nutrition and Food Technology “José Mataix”, CIBM, University of Granada, Armilla, 18016 Granada, Spain; 3Instituto de Investigación Biosanitaria ibs (IBS), 18012 Granada, Spain

**Keywords:** probiotics, *Bacillus*, antimicrobial effect, in vitro methods, in silico methods

## Abstract

The variable taxa components of human gut microbiota seem to have an enormous biotechnological potential that is not yet well explored. To investigate the usefulness and applications of its biocompounds and/or bioactive substances would have a dual impact, allowing us to better understand the ecology of these microbiota consortia and to obtain resources for extended uses. Our research team has obtained a catalogue of isolated and typified strains from microbiota showing resistance to dietary contaminants and obesogens. Special attention was paid to cultivable *Bacillus* species as potential next-generation probiotics (NGP) together with their antimicrobial production and ecological impacts. The objective of the present work focused on bioinformatic genome data mining and phenotypic analyses for antimicrobial production. In silico methods were applied over the phylogenetically closest type strain genomes of the microbiota *Bacillus* spp. isolates and standardized antimicrobial production procedures were used. The main results showed partial and complete gene identification and presence of polyketide (PK) clusters on the whole genome sequences (WGS) analysed. Moreover, specific antimicrobial effects against *B. cereus, B. circulans, Staphylococcus aureus, Streptococcus pyogenes, Escherichia coli, Serratia marcescens, Klebsiella* spp., *Pseudomonas* spp., and *Salmonella* spp. confirmed their capacity of antimicrobial production. In conclusion, *Bacillus* strains isolated from human gut microbiota and taxonomic group, resistant to Bisphenols as xenobiotics type endocrine disruptors, showed parallel PKS biosynthesis and a phenotypic antimicrobial effect. This could modulate the composition of human gut microbiota and therefore its functionalities, becoming a predominant group when high contaminant exposure conditions are present.

## 1. Introduction

The human gut microbiota could be considered as a new source for the identification and isolation of multiple microorganisms producing bioactive compounds and enzymes of interest such as biopolymers, antimicrobials notably demanded by the food, health, and several biotechnological industries [1,2]. Identifying the composition of cultivable gut microbiota has always been a challenge due mainly to the requested anaerobic conditions [3]. Efforts in simulating these harsh culture conditions allow isolating potential NGP [4] and even a variety of taxonomy bacterial groups which were also tolerant to xenobiotics or obesogens [5] followed by characterization through 16S rRNA gene sequencing.

Microbiome compositional consortia are variable in each individual [6,7]. Culturing methods and directed-culturomics for isolating specific microorganisms deserve special attention. Thus, the genus *Bacillus* belonging to a predominant microbiota phylum, Firmicutes, is differentially present and its species are capable of synthesizing a wide variety of bioactive compounds and enzymes of interest for their potential technological applications in health and the modern food biotechnological sectors [8]. Several *Bacillus* species have also been considered as probiotics [9,10]. *Bacilli* taxa, concretely *Lactobacillus* and *Bacillus* genera in microbiota seem to play a role on the ecology of predominant groups present on individual microbiota in obesity and metabolic disorders as compiled in human clinical trials (Table 1). The potential impact on the other circumscribed taxa groups could be driven by antimicrobial substances released by the *Bacilli* taxa, such as bacteriocins, PKs, lipopeptides, etc. [11,12].

Bisphenols are considered as microbiota disrupting chemicals (MDC) [5] and their presence in humans has been confirmed by detecting them in human biospecimens: feces, serum, urine, saliva, hair, tissue and blood [23,24]. Bisphenol A (BPA) is used in manufacturing polycarbonate and epoxy resins for food consumer products and packages. There is also cumulative exposure from contaminating soils, aquatic environments, drinking water, air and dust particles [25]. The estrogen activity alteration is the most widely studied effect of BPA and analogues, enhancing endocrine disruptor activities [26]. Moreover, some studies have shown obesogenic effects through microbiota dysbiosis [27], fat cell development, and lipid accumulation [28]. There are several regulations enforced concerning the hazards of Bisphenol A, as derivative of polycarbonates plastics and epoxy resins, used in food contact materials, toys, or other products. In order to protect the consumers from cumulative exposure, the tolerable daily intake (TDI) for BPA is permanently re-evaluated according to new toxicity data through specific international projects, such as U.S. National Toxicology Program (CLARITY-BPA program) [29] or European Food Safety Authority (EFSA) comprehensive re-evaluation of BPA exposure and toxicity [30].

Moreover, commensal microorganisms isolated from human microbiota could in general fulfill the criteria of safety assessment and the status of Qualified Presumption of Safety (QPS) [31,32]. Similarly, most *Bacillus subtilis* cluster species are considered QPS [33] and they are increasingly marketed as products [34]. Conversely, *Bacillus cereus* cluster species can be also present in the gut microbiota, but they are not considered as QPS [34,35].

Next-generation sequencing (NGS) platforms and WGS of microorganisms have enlarged the molecular comparison knowledge on the gene collection for encoding enzymes, and better taxonomy has supported appropriate classification. Moreover, specific WGS gene description is needed to consider the food and feed safety aspects of microbiota cultivated strains [35].

Genome mining tools and phenotypic analysis are complementary approaches to predict and demonstrate the production of active secondary metabolites such as antimicrobial products from *Bacillus* species [36]. Genome mining revealed the potential for known and novel PKs extensively in *Bacillus* (Figure 1). Moreover, based on the prediction of the general architecture, novel clusters were identified in novel *Bacillus* spp. variants. In addition, more recent in silico and bioinformatics approaches seem to be successful to find and verify the microbial potential to produce valuable enzymes for biotechnological applications [36].

The main objective of the present study was to determine the antimicrobial effects of catalogue of microorganisms isolated from human gut, by applying directed-culturing methods after the addition of endocrine disruptor chemicals. Taxa groups of isolated bisphenol A (BPA)-degrading *Bacillus* spp. will be analyzed by with in vitro assays to demonstrate the bioactive substances released against commensals and critical pathogens according to the World Health Organization (WHO). Moreover, genome mining and in silico tests will be used for disclosing the genes responsible for antimicrobial production and its enzymatic pathways.

## 2. Materials and Methods

### 2.1. Microbiota Sampling Bank and Directed Culturing Approach

Ten isolates from fecal human microbiota collections of 0–1 year old infants (Isolates B-Project INFABIO) appropriately maintained at −80 °C underwent a directed culturing approach using 0.5 g of the fecal specimen in 1.5 mL of Brain Heart Infusion or Man Rogosa and Sharpe (BHI/MRS) broths, adding different concentrations of BPA (0.5, 10, 20, and 50 ppm), in order to search tolerant and/or potentially BPA biodegrading microorganisms, incubation for 72 h. Further serial dilutions and spreading onto BHI/MRS solid media plus incubation under aerobic and anaerobic conditions (anaerobic jars anaerocult^®^) at 37 °C over 72 h were applied. BPA-tolerant colonies with distinguishing features were isolated as pure culture for subsequent morphological, phenotypic, and genotypic identifications: bacterial cell counts, gram staining, spore staining, capsule staining, catalase activity, oxidase, and motility tests.

### 2.2. BPA Microbiota Tolerance Testing

BPA biodegradation microbiota capacity was tested directly adding BPA to the human fecal samples. The specimens were exposed to 25 ppm concentration of BPA at 30 °C during 72 h. BPA was measured in the extracts and supernatants through Liquid chromatography–mass spectrometry (LC-MS/MS) system for BPA quantification. Chemicals, reagents, instrumentation, and software for bisphenols determination were provided by CIC services under validated procedures previously described by García-Córcoles et al. [38].

### 2.3. Culturing- Isolation of Bacillus Catalogue

A common approach to isolate *Bacillus* strains from microbiota has been pursued in our research team [39]. For this study, ten isolates from fecal human microbiota collections of 0 to 1 year old infants (Isolates B-Project INFABIO) and 6–8 year-old children (Isolates C-Project OBEMIRISK) were obtained by a serial dilution method, with exposure to different BPA concentrations (0.5, 10, 20, and 50 ppm) over 72 h and further spreading in BHI/MRS media incubated under aerobic and anaerobic conditions (anaerobic jars anaerocult^®^) at 37 °C. The BPA-tolerant bacterial colonies with distinguishing features were isolated as pure culture for subsequent morphological, phenotypic, and genotypic identifications: bacterial cell counts, gram staining, spore staining, capsule staining, catalase activity, oxidase, and motility tests.

### 2.4. Genomic DNA Extraction, Taxonomy Identification and Phylogenetic Analysis

Genomic DNA was extracted using DNeasy columns (Qiagen^®^, Hilden, Germany) following the manufacturing instructions. The isolated DNA was quantified using Nanodrop (Thermo Scientific^®^ Waltham, MA, USA) and biophotometer (Eppendorf^®^ D30). The quality of DNA was monitored through gel electrophoreses. Complete 16S RNA gene sequencing of selected bacterial strains was done by Sanger method (Institute of Parasitology and Biomedicine “López-Neyra” IPBLN Service). Forward and reverse sequences were provided separately. Reverse sequence was converted to complementary sequence with Chromas Pro 2.0 software (Technelysium Pty Ltd., Tewantin, Australia). Sequences were examined for maximum homology against GenBank using National Center for Biotechnology Information NCBI’s BLASTn program. The collection and comparison of complete 16S rRNA gene sequences were performed using the Ezbiocloud platform [40].

### 2.5. Enzymes Tests

Relevant enzymatic production assays were carried out to verify the potential of gut microbiota strains to synthetize relevant enzymes in the biotechnological and industrial context. Starch, carboxymethylcellulose, inulin, tween 20 and 80, and DNase supplemented media were used to determine the degradation of different substrates according to complementary methodologies [41,42,43,44,45,46].

### 2.6. Antimicrobial In Vitro Tests

Antimicrobial activity was tested by agar well diffusion method. Under Joint FAO/WHO Expert Committee on Food (JECFA) procedures [47] and the study carried out by Powthong & Suntornthiticharoen [48], nine different bacteria were used as indicators to verify the antimicrobial capacity of the *Bacillus* spp. isolated from the gut microbiota. To determine the synthesis of antimicrobial compounds, several isolated strains were selected according to preliminary antimicrobial tests and the main taxonomy groups: strains close/represented by rB1 (*Bacillus* sp. AM1), strains close/represented by rB3 (*Bacillus siamensis* (KCTC 13613)), strains close/represented by rB7 (*Bacillus cereus* (AFS039342)). Plates with 20 mL of Müller-Hinton agar were prepared and test microorganisms used as indicators: *Bacillus cereus, Bacillus circulans, Staphylococcus aureus, Streptococcus pyogenes, Escherichia coli, Serratia marcescens, Klebsiella* spp., *Pseudomonas* spp., and *Salmonella* spp., were adjusted to a cell density of 0.5 on the McFarland scale in sterile 0.85% NaCl solution. The data were expressed as mean of the three replicates. Tests were done spreading the indicator microbial strains over the surface of the Müller-Hinton agar using sterile cotton swab. Inside six mm diameter oxford wells generated in agar, 20 µL of antibiotic producing bacteria extract was added. Standards appropriate positive controls (ampicillin, gentamycin, and streptomycin at 10 µg) and negative/blank (sterile media/ethanol) were used. The plates were incubated at 37 °C for 24 h and the inhibition zones were measured.

### 2.7. Genome Data Mining and Analysis –PKs Genes and Clusters

#### 2.7.1. Genome Mining Tools for PKs Gene Searching

In order to discover the presence of secondary metabolites, several bioinformatics tools were used to perform genome mining. A data retrieving software has been specifically computed using Pascal programming language to obtain the PKs enzymes ID and the corresponding Loci from the genomes.

Type strain genomes from the closest species isolated were retrieved from NCBI Genome Data Bank in GenBank file format in order to list the proteins that they were able to potentially produce.

A more detailed prediction of the clusters was performed by checking the downstream and upstream genes of those involved in PKs synthesis using NCBI genome map viewer [49].

#### 2.7.2. Prediction of Polyketides in WGS of *Bacillus* sp. AM1 Isolated from Microbiota

The identification of PKs gene cluster was carried out by the analysis of the WGS of *Bacillus* sp. AM1, GenBank CP047644.1, following the same approach explained above.

## 3. Results and Discussion

### 3.1. BPA-Tolerant Microorganisms Isolated from Human Gut Microbiota

#### 3.1.1. BPA Microbiota Metabolization Capacities

The microbiota composition of each fecal sample was specific and contributed differentially to the biodegradation of BPA exposure levels (Figure 2). Each fecal sample (340, 349, and 437) showed a differential ability to eliminate BPA due to its taxa compositional and functional characteristics, showing sample 340 a maximum percentage of BPA degradation of 89.3% while sample 349 degraded 76% and 437 was able to eliminate 21% of the BPA concentration. Previous studies have shown the same effects in the environment [50], where they observed that different microbial communities presented a specific elimination rate dependent on their composition.

Cumulative exposure to a wide range of xenobiotics, such as BPA and its analogues, affects the microbiota diversity possessed by each individual, causing a selection of bacteria strains to populate the gut, and consequently modify its equilibrium through MDC [5]. This dysbiosis has been proven to be responsible for well-known diseases, such as obesity, diabetes, and even some hormonal-related cancers. Therefore, identification of the triggered main taxa variations and their functions remains a challenge. Moreover, the appropriate use of probiotics [50,51,52] or search for NGP to mitigate or reverse these dysbiosis are crucial [53,54]. A directed culturing approach allow us to select tolerant bacteria and mimic an ecological environment to understand better the impact of the specific enriched communities and their capacities to impact the taxa microbiota colonization.

#### 3.1.2. Catalogue of BPA-Tolerant *Bacillus* spp. Isolated from Human Microbiota

Isolation and identification of BPA-tolerant *Bacillus* spp. strains from microbiota samples were successfully performed with the different BPA concentrations plates (0.5; 10; 20 and 50 ppm). Out of these 11 isolates analyzed, the closest species by complete gene 16S rRNA sequence were *B. amyloliquefaciens, B. siamensis, B. velezensis, B. nematocida, B. cereus,* and *B. pacificus* (Table 2).

Data obtained by parallel experimental work showed a BPA directed human fecal culturing catalogue that contained different BPA tolerant species from the following genera and percentages: *Enterococcus* 28%, *Bacillus* 27%, *Staphylococcus* 10%, *Escherichia* 8%, *Clostridium* 5%, and *Lactobacillus* 4% (data not shown). Representing *Bacilli* taxa (*Bacillus* and *Lactobacillus*) was a major taxa with approximately a 30% of BPA tolerant isolated strains from microbiota samples, which corroborates the predominant presence of these genera being able to overcome the impact of xenobiotics, such as BPA, as previous assays showed [39].

In line with these results, interesting properties and uses are specifically described for *Bacillus* spp. Recently, several *Bacilli* strains have been extensively proposed for use as human and animal probiotics [55,56]. Most of the species used belong to *Bacillus subtilis* and *Bacillus amyloliquefaciens* groups and special attention should be paid to the food and clinical studies with strains that showed special enzyme capacities [57] or those able to modulate and mitigate pathophysiological disorders [58].

#### 3.1.3. Taxonomical and Phylogenetic Clustering

The phylogenetic tree based on complete 16S rRNA gene of *Bacillus* strains isolated from microbiota treated with BPA grouped the clusters to *B. subtilis, B. amyloliquefaciens*, *B. velezensis, B. siamensis, B.cereus,* and *B. pacificus* (Figure 3). The two main clustering of closely related *Bacillus* strains belong to *B. subtilis* and *B.amyloliquefaciens* taxonomic group (green) and *B. cereus* group (yellow). Three representative strains (rB1, rB3, and rB7) were further processed by bioactive compounds production tests. They were organized as follows: rB1 represented B1, B4, B5, B6, B7, B8, B9, and B9.2; rB3 represented B2 and B3; rB7 represented B7 and B12.

The strains isolated in the present work were clustered in the two main groups: *B. subtilis*–like (non-pathogenic) [59] and *B. cereus*-like (pathogenic) [60], as shown in Figure 3, however the pathogenicity features are strain-specific dependent. The work approach is based on potential uses and predictive data analysis, but for further commercial uses, a safety assessment should be performed for each strain, to demonstrate that they do not pose any safety and/or pathogenicity concerns. The battery of tests usually requested is: antibiotic resistance test no greater than existing regulatory cutoffs against clinically important antibiotics, incapacity to induce hemolysis or produce surfactant factors, and the absence of virulence or toxigenic activity in vitro.

### 3.2. Analysis of Bioactive Compounds Production Capacities

#### 3.2.1. Enzymatic Activity Tests

*B. subtilis, B. amyloliquefaciens,* and *B. licheniformis* have been used as bacterial resources in the industrial context for the production of a wide range of enzymes and bioactive compounds for decades. *Bacillus* sp. AM1 and other strains belonging to *Bacillus* genus have shown remarkable hydrolytic enzyme capacity (Table 3), being related to the performance of key roles in several biotechnological and many manufacturing processes [61,62,63].

#### 3.2.2. Antimicrobial Activity Tests

The results obtained from antimicrobial experimental tests carried out with the representative isolated microorganisms from different taxonomic clusters confirmed the ability of the strains B1 and B3 to inhibit Gram-negative and Gram-positive bacteria (Table 4).

Preliminary results grouped the strains according to their capacity of antibiotic production with very similar inhibiting zone value, which were also in agreement with the main taxonomic clusters. rB1 represented B1, B4, B5, B6, B7, B8, B9, and B9.2; rB3 represented B2 and B3; rB7 represented B7 and B12.

rB1 and rB3 strains were found to be antagonistic against Gram-positive *Bacillus cereus*, *Bacillus circulans*, *Staphylococcus aureus, Streptococcus pyogenes* (diameter of zone of growth inhibition 10–17 mm) and also against Gram-negative food-borne pathogenic bacteria *Serratia marcescens,*
*Escherichia coli*, *Salmonella*, and *Klebsiella pneumoniae* (diameter of zone of growth inhibition 10–20 mm). Conversely, the strains rB7 did not show any production of antimicrobial effects.

Minimum inhibitory concentration (MIC) values were similar to those resultant of other polyketides antimicrobial effects previously described, being significant differential and higher the effects found against *Klebsiella* [64]. Therefore, the search for a putative biosynthetic pathway of the *pks* gene product proceeded after the validated molecular antimicrobial attributions.

### 3.3. WGS Data Mining and In Silico Analysis

#### 3.3.1. WGS Mining in Type Strains

The bioinformatics analysis carried out on the type strains of closest species identified as cultivable *Bacillus* species from microbiota showed specific enzymes involved in PKs biosynthesis (Table 5). The genome mining identified the clusters with the genomes from closest homologue type strains available in the database. Bioinformatic tools and Pascal ad hoc software allowed the exhaustive analysis of genomes making it a powerful prediction tool.

According to the results, *Bacillus*
*amyloliquefaciens, B. siamenensis, B. velezensis, B. subtilis* and *B. atrophaeus* harbor almost complete *pks* genetic macroclusters for the production of polyketides. While *B. licheniformis, B. cereus, B. pacificus*, and the *probiotics B. clausii, B. coagulans* did not contained the PKs loci. The antimicrobial effects of polyketides are site colonization specific and the strains are scarcely used for health biotechnological interests [65]. Moreover, the ecological impact of these antimicrobial substances on the gut microbiota composition may have a huge impact, beyond the modification and control of the colonization of commensals and pathogenic bacteria, e.g., to cause weight gain effects in humans as well as in animals [66].

#### 3.3.2. WGS Representative *Bacillus* sp. AM1 from Microbiota: Genome Mining Data

From the analysis of the specific *Bacillus* sp. AM1 WGS, the cluster genes and enzymes related to PKs biosynthesis were identified (*bae*, *mln*, and *dfn*) and they were related to the production of bacillaene, and two other polyketides macrolactin and difficidin.

This complex microbial ecosystem seems to be enriched in new bacterial strains belonging to *Bacillus* genus that produce PKs with a wide range of applications in the current biotechnological context. Among these applications, PKs stand out for their antimicrobial capacity against certain bacterial species. Therefore, further identification through bioinformatics tools and experimental data will confirm the functionality of these bioactive substances.

Advances in NGS and in silico tools allow to perform an appropriate screening of genes of concern or interest in microbiota, such as antimicrobial resistance genes and the capacity of antimicrobial production of cultivable isolates WGS. A better understanding of the microbiota ecology, driven by the bioactive compounds released by its components, will lead to better clinical interventions. Antimicrobials naturally synthetized by gut microorganisms are mainly described as bacteriocins [12]. However, it is important to consider other molecules acting as antimicrobial as polyketides. Isolation and elucidation of PKs structures by nuclear magnetic resonance (NMR) methods are limited by the concentration needed for analysis [67]. Thus, it is possible to predict the types of PKs and their variants, as showed for Bacillales [37]. Genome mining performed in the present study allowed BLAST driven search for predicted PKs clusters. Pascal ad hoc software analysed the type strain genomes making it a powerful prediction tool. Similarly, another useful prediction tool could be used as nonribosomal peptide-synthetase NRPS/PKs substrate predictor [68].

Importantly, *Bacillus* and specific WGS genes description is needed to verify the safety assessment of different strains if they are proposed to be used in food or feed chain [70]. Moreover, the safety of a beneficial microbe or probiotic strain must be sufficiently characterized by high-throughput technologies, safe for the intended use, and assessed through pathogenicity, immunotoxicity, and colonization, in addition to its antibiotic resistance profile [71]. However currently, there is no consensus or standardization for the interventional use of probiotics [72]. In addition to general guidelines for the qualification of the QPS, European Food Safety Authority (EFSA) made a supplementary requirement for *Bacillus* species other than the *Bacillus cereus* group, where a cytotoxicity test should be performed to determine whether the strain produces high levels of non-ribosomal synthesised peptides. One of the criteria for strains to fulfill and meet the requirements for QPS and generally recognized as safe (GRAS) standards is antimicrobial activity and the absence of antimicrobial resistance genes as a possible safety concern against critically important antimicrobials (CIAs) or highly important antimicrobials (HIAs), which might eventually be transferred via horizontal gene transfer to pathogenic bacteria during food manufacture or after consumption [33,73]. According to the general guidelines for the qualifications of the QPS, unless the strain qualifies for the QPS approach or belongs to a taxonomic unit, known not to produce antimicrobials relevant to use in humans and animals, assessment should be made to determine the inhibitory activity of culture supernatants against reference strains, known to be susceptible to a range of antibiotics and the inhibitory substance [47]. A slight adjustment has been made for the production strains, which have to demonstrate the absence of carry-over into the final product together with the exact phase of the industrial scale manufacturing process, and whether any CIAs or HIAs are used during the manufacturing of the product, to determine compatibility with other additives showing antimicrobial activity and, furthermore, possible co-/cross-resistance [35].

## 4. Conclusions

*Bacillus* strains isolated from human gut microbiota, and taxonomically closest to the safely qualified *B. subtilis* and *B. amyloliquefaciens* groups, became cultivable predominant taxa when high bisphenol exposure conditions were tested. In parallel, these strains harbored PKS molecular gene biosynthetic loci and showed phenotypic antimicrobial effects. Therefore, they might be proposed as beneficial microorganisms with molecular features that would contribute to modulate the ecological taxa composition and functionality of human gut microbiota. Intervention studies will be further needed to demonstrate the ability to recover from microbiota dysbiosis, triggered by high MDC exposure diets and lifestyles, towards eubiosis and healthier status.

## 5. Patents

IPR-823 Application in progress.

## Figures and Tables

**Figure 1 microorganisms-09-01615-f001:**
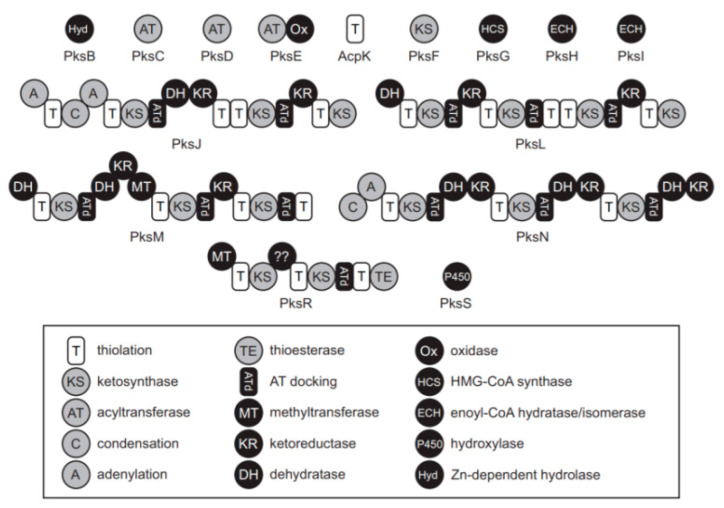
Conserved PKs proteins and functions in *Bacillus* modified from Straight et al. [37].

**Figure 2 microorganisms-09-01615-f002:**
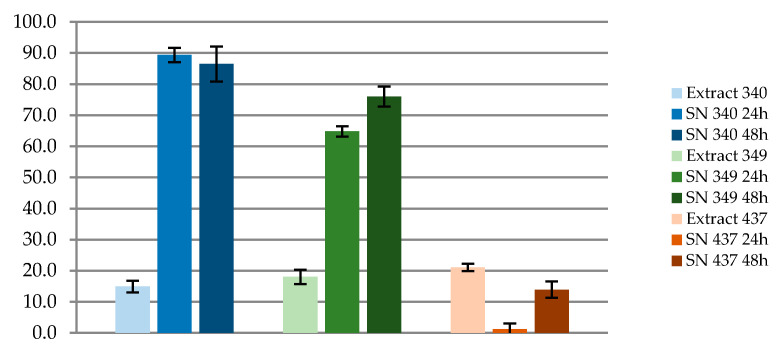
BPA relative percentage of degradation by human fecal specimens. (LC-MS/MS) system was used for BPA quantification; SN: Supernatant.

**Figure 3 microorganisms-09-01615-f003:**
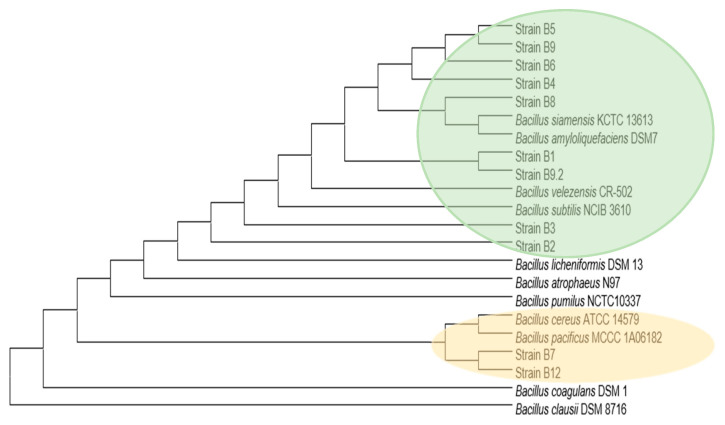
Phylogenetic tree based on gene sequences of isolated gut microbiota strains. The tree was obtained by applying Neighbor-Join and BioNJ algorithms to a matrix of pairwise distances estimated using the Maximum Composite Likelihood (MCL) approach and Kimura 2-parameter model. The species and strain names are shown. Bootstrap values shown after 1000 resamplings. Main clusters are highlighted: in green close to *B.subtilis* group and yellow close to *B.cereus* group.

**Table 1 microorganisms-09-01615-t001:** *Bacilli taxa* modifications from clinical trials of metabolic related diseases.

Reference	Clinical Trials—Disease /Sample Size and Clinical Traits	Taxa Modifications
[13]	OB; *n* = 192; HC *n* = 25; OW *n* = 22; OB *n* = 145	↑ ***Bacillus*** in OW and OB
[14]	OB, AN; *n* = 49; HC *n* = 20; OB *n* = 20; AN *n* = 9	↑ ***Lactobacillus*** in OB
[15]	T2D; *n* = 36; HC *n* = 18; T2D *n* = 18	↑ ***Lactobacillus*** in T2D
[16]	T2D, OB; *n* = 60; HC *n* = 20; Obese-T2D *n* = 40	↑ ***Bacillus sporothermodurans*** in OB-T2D
[17]	T1D, T2D; *n* = 110; HC *n* = 40; T2D *n* = 49; T1D *n* = 21	↑ ***Lactobacillus*** in T1D and T2D
[18]	NAFLD; *n* = 126; HC *n* = 83; NAFLD *n* = 43	↓ ***Lactobacillus*** in NAFLD
[19]	NAFLD; *n* = 67; HC *n* = 37; NAFLD *n* = 30	↑ ***Lactobacillaceae*** in NAFLD
[20]	NAFLD; *n* = 60; HC *n* = 30; NAFLD *n* = 30	↑ ***Lactobacillus*** in NAFLD
[21]	NAFLD, OB; *n* = 73; HC *n* = 20; OB-NAFLD *n* = 36; OB-non-NAFLD *n* = 17	↑ ***Bacilli*** in OB-NAFLD ↑ ***Lactobacillus*** in non-NAFLD
[22]	MetS; *n* = 655; Monozygotic twins *n* = 306; Dizygotic twins *n* = 74; Siblings *n* = 275	↑ ***Lactobacillus*** in MetS

AN: anorexia nervosa; HC: healthy control; MetS: metabolic syndrome; NAFLD: non-alcoholic fatty liver disease; OB: obese; OW: overweight; T1D: type 1 diabetes; T2D: type 2 diabetes. ↑ Increasements.

**Table 2 microorganisms-09-01615-t002:** *Bacillus* isolates from human microbiota and 16S rRNA complete gene homology description.

Microbiota Isolates	Closest Taxa—[Strain] Best Hit	bp Position 16S rRNA	Query Cover (%)	Identity (%)	Accession Number
B1	*Bacillus siamensis* [LRM10-3D]	15,030	100	100	MT645306.1
	*Bacillus velezensis* [XC1]		100	100	MT649755.1
B2	*Bacillus velezensis* [CR-502]	1483	95.4	99.14	AY603658
B3	*Bacillus siamensis* [KCTC 13613]	1490	100	98.00	AJVF01000043
B4	*Bacillus siamensis* [KCTC 13613]	1515	100	99.66	AJVF01000043
	*Bacillus nematocida* [B-16]		100	99.73	AY820954
	*Bacillus amyloliquefaciens* [DSM7]		100	99.52	FN597644
B5	*Bacillus siamensis* [KCTC 13613]	1516	100	98.91	AJVF01000043
	*Bacillus nematocida* [B-16]		100	98.98	AY820954
	*Bacillus velezensis* [CR-502]		95.4	99.22	AY603658 FN597644
	*Bacillus amyloliquefaciens* [DSM7]		100	98.78	
B6	*Bacillus velezensis* [CR-502]	1504	95.4	99.93	AY603658
B7	*Bacillus cereus* [AFS039342]	1510	100	99.39	NUMR01000072
	*Bacillus pacificus* [NCCP 15909]		100	99.34	CP041979.1
B8	*Bacillus velezensis* [CR-502]	1520	95.4	99.93	AY603658
B9	*Bacillus velezensis* [CR-502]	1499	95.4	99.22	AY603658
B9.2	*Bacillus siamensis* [KCTC 13613]	1499	100	99.52	AJVF01000043
	*Bacillus nematocida* [B-16]		100	99.59	AY820954
	*Bacillus amyloliquefaciens* [DSM 7]		100	99.39	FN597644
B12	*Bacillus cereus* [AFS039342]	1543	100	99.39	JMQC01000008
	*Bacillus pacificus* [NCCP 15909]		99.0	99.35	CP041979.1

**Table 3 microorganisms-09-01615-t003:** Enzymatic activity in gut microbiota isolates.

Enzyme Test	Microbiota Isolates		
	rB1	rB3	rB7
Starch	+	++	++
Carboxymethylcellulose	-	-	-
Inulin	+	-	+
Tween 80	-	-	-
DNase	++	-	-

**Table 4 microorganisms-09-01615-t004:** Antimicrobial activity of BPA-tolerant human gut microbiota isolated strains.

Target Indicator Bacteria	Strains rB1	Strains rB3	Strains rB7
	Diameter of inhibitory zone (mm) ± SD ^1^		
*Bacillus cereus*	15 ± 0	17 ± 0	-
*Bacillus circulans*	13 ± 0	14.3 ± 1.2	-
*Staphylococcus aureus*	11.7 ± 0.6	10 ± 0	-
*Streptococcus pyogenes*	15 ± 0	13.3 ± 0.6	-
*Serratia marcescens*	17 ± 0	15.3 ± 1.5	-
*E. coli*	15 ± 0	13.3 ± 0.6	-
*Salmonella*	11 ± 0	10 ± 0	-
*Klebsiella*	20 ± 0 *	15 ± 0 *	-
*Pseudomonas*	-	-	-

^1^ Values are mean diameter of inhibitory zone (mm) ± SD of three replicates. The diameter of well (6 mm) was included. (-) Diameter of inhibitory zone <7 mm considered as no antimicrobial activity. * Significant values compared to theroretical values from *B. subtilis* polyketides [64].

**Table 5 microorganisms-09-01615-t005:** Gene-encoding and corresponding enzymes involved in Polyketide biosynthesis in WGS of Type strain of *Bacillus spp*.

Enzyme	Enzyme description EC number	B. amyloliquefaciens WF02T NZ_CP053376	B. siamenensis SCSIO 05746T NZ_CP025001	B. velezensis CBMB205T NZ_CP011937	B. subtilis 168T NC_000964	B. atrophaeus BSST NZ_CP007640	B. sp-AM1 B1T CP047644.1)
PksA	Hypothetical protein/EC:3.1.2.6	WP_024085315.1 174131..1741526	WP_060962748.1 2494188..2494397	WP_032874955.1 2222103..2222312	NP_000389590.1 1782713..1783390	WP_013390522.1 1165636..1167084	1787442..1787651 QHJ03379.1
-	Hypothetical protein/EC:3.1.2.6	WP_024085326.1 1816193..1816555	WP_016936035.1 2419160..2419522	WP_007410383.1 2146808..2147170	YP_0009513956.1 1783500..1783766	WP_003328852.1 1167393..1167932	*-*
Regulator	TetR family transcriptional regulator C terminal	-	-	-	NP_000389589.1 1781906..1782523	WP_003328851.1 1168054..1168644	*-*
PksB	MBL fold metallo hydrolase/ EC: 2.3.1.39	WP_024085316.1 1742160..1742837	WP_060962747.1 2492787..2493464	WP_032874957.1 2220496..2221173	YP_0009513956.1 1783500..1783766	WP_003328850.1 1168942..1169619	1788295..1788972 QHJ03380.1
PksC	ACP S malonyltransferase/ EC:2.3.1.51	WP_014305029.1 1743152..1744021	WP_060962746.1 2491603..2492472	WP_032874959.1 2219312..2220181	NP_000389591.1 1783763..1784629	WP_003328849.1 1170013..1170879	1789287..1790156 QHJ03381.1
PksD	Acyltransferase domain containing protein/EC: 2.3.1.39	WP_003154101.1 1744158..1745132	WP_060962745.1 2490494..2491468	WP_032874961.1 2218201..2219175	NP_000389592.2 1785133..1786107	WP_003328847.1 1171417..1172382	1790293..1791267 QHJ03382.1
PksE	ACP S malonyltransferase/ EC:1.3.1.9 and 1.3.1.10	WP_003154100.1 1745134..1747374	ID Not found 2488250..2490492	WP_032874963.1 2215959..2218199	NP_000389593.3 1786104..1788407	WP_003328846.1 1172389..1174752	1791269..1793509 QHJ03383.1
AcpK	Acyl carrier protein/EC:2.3.3.10	WP_003154099.1 1747440..1747688	WP_060962743.1 2487934..2488182	WP_012117592.1 2215645..2215893	NP_00570904.1 1788469..1788717	WP_003328845.1 1174891..1175139	1793575..1793823 QHJ03384.1
PksF	Polyketide beta ketoacyl:ACP synthase/EC: 4.2.1.17	-	-	-	NP_000389594.2 1788695..1789942	WP_003328844.1 1175117..1176364	*-*
PksG	Hydroxymethylglutaryl CoA synthase family/EC: 4.2.1.17	WP_003154098.1 1747740..1749002	WP_060962742.1 2486620..2487882	WP_032874965.1 2214331..2215593	NP_000389595.2 1789943..1791205	WP_010788667.1 1176364..1177626	1793875..1795137 QHJ03385.1
PksH	Enoyl CoA hydratase/isomerase	WP_024085319.1 1748999..1749772	WP_060962741.1 2485850..2486623	WP_032874967.1 2213561..2214334	NP_000389596.1 1791193..1791972	WP_087941777.1 1177614..1178390	1795134..1795907 QHJ03386.1
PksI	enoyl CoA hydratase/isomerase family protein	WP_003154094.1 1749782..1750531	WP_060962740.1 2485091..2485840	WP_003154094.1 2212802..2213551	NP_000389597.2 1792012..1792761	WP_003328841.1 1178438..1179184	1795917..1796666 QHJ03387.1
PksJ	Non ribosomal peptide synthetase	WP_024085320.1 1750571..1765525	WP_060962739.1 2470129..2485062	WP_032874969.1 2197814..2212762	NP_000389598.3 1792806..1807937	WP_013390525.1 1179247..1194429	1796706..1811657 QHJ03388.1
PksM	SDR family NAD(P) dependent oxidoreductase EC:1.6.5.2	WP_165869029.1 1765509..1778951	WP_167388675.1 2456724..2470145	WP_162859398.1 2184400..2197830	NP_000389601.3 1821553..1834341	WP_013390526.1 1194431..1208248	1811659..1825086 QHJ03389.1
PksM	SDR family NAD(P) dependent oxidoreductase/EC:1.6.5.2	WP_024085322.1 1778969..1789513	WP_101605493.1 2446202..2456707	WP_032874973.1 2173847..2184382	NP_000389602.3 1834409..1850875	WP_013390527.1 1208267..1221238	1825104..1835639 QHJ03390.1
PksN	Non ribosomal peptide synthetase	-	WP_101605492.1 2429908 2446212	WP_032874975.1 2157559 2173857	NP_000389604.2 1850890 1858521	WP_087941783.1 1221318..1237793	1835629..1851930 QHJ03391.1
PksR	Polyketide synthase dehydratase domain/EC:2.1.1.-	WP_024085324.1 1805818..1813275	WP_060962735.1 2422440..2429894	WP_032874977.1 2150088..2157545	NP_000389600.3 1807921..1821537	WP_003328830.1 1237809..1245533	1851944..1859401 QHJ03392.1
PksS	Cytochrome P450/EC:1.14.14.-	WP_024085325.1 1813410..1814621	WP_060962734.1 2421090..2422301	WP_032875233.1 2148742..2149953	NP_000389605.2 1858566..1859783	WP_003328829.1 1245647..1246888	1859536..1860747 QHJ03393.1

*B. licheniformis* (strain ATCC 14580)^T^; NC_006270 PKs Loci was not found; *B. cereus (strain B4264)* NC_011725 PKs Loci was not found; *B. pacificus* (strain R1) NC_NJQG01000001 Loci was not found; *B. clausii* (strain 7520-2 contig00001)^T^ NZ_NPBN01000001 PKs Loci was not found; *B. coagulans* (B4099 NODE_1)^T^ NZ_LQYI01000001 PKs Loci was not found; *B. nematocida* (strain B-16^T^) No WGS is available—Analysis PKS Loci was not applicable [69].

## Data Availability

Not applicable.

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
