# Peer review of "Antimicrobial Effects of Potential Probiotics of Bacillus spp. Isolated from Human Microbiota: In Vitro and In Silico Methods"

_microorganisms, 2021, doi:10.3390/microorganisms9081615_

Round 1

Reviewer 1 Report

The authors presented an interesting new perspective on the antimicrobial effects of potential probiotics of Bacillus spp. isolated from human microbiota, based on in vitro and in silico approaches. 

I found the approach they used very interesting and innovative, the research design appropriate, and the methods adequately described.

However, before manuscript's publication, I strongly recommend:

  • overall, a revision of the English (I found some Spanish terms, like "Tabla 1", line 56). 
  • A revision of the use of the terms "microbiota" vs "microbiome".
  • To avoid, when possible, terms like microbes/microbial as de facto the manuscript is on a subset of microbes, i.e. bacteria.
  • To add references when using concepts like "microbiome compositional consortia" (line 47)
  • To use of italics for species names (see lines 262-270)
  • To imprve of the introduction by explaining why the authors focus the attention on bisphenol. In particular, I suggest to add a paragraph with references to policy regulations about (mis)use of bisphenols, as background information for the reader.
  • To add a conclusion paragraph/section, as the Abstract has a paragraph about "our conclusions" which is then not expanded in the manuscript. In this section, I recommend also to add considerations about what is missing to consider the described effects as "concrete", especially in the context of healthy both diet and life style.
  • To change of colours/patterns of bars represented in Figure 2, which I found difficult to be read. 

Author Response

REVIEWERS COMMENTS:

AUTHORS ANSWERS

REVIEWER 1:

The authors presented an interesting new perspective on the antimicrobial effects of potential probiotics of Bacillus spp. isolated from human microbiota, based on in vitro and in silico approaches. I found the approach they used very interesting and innovative, the research design appropriate, and the methods adequately described.

Thank you very much for the positive evaluation of our manuscript.

However, before manuscript's publication, I strongly recommend:

  • overall, a revision of the English (I found some Spanish terms, like "Tabla 1", line 56). 

DONE as suggested. English native revision has been done for the whole new version of the manuscript

  • A revision of the use of the terms "microbiota" vs "microbiome".

DONE as suggested.

  • To avoid, when possible, terms like microbes/microbial as de factothe manuscript is on a subset of microbes, i.e. bacteria.

DONE as suggested.

  • To add references when using concepts like "microbiome compositional consortia" (line 47).

DONE as suggested. The following references have been added:

Adair KL, Douglas AE. Making a microbiome: the many determinants of host-associated microbial community composition. Curr Opin Microbiol. 2017;35:23–9.

Diakite A, Dubourg G, Dione N, Afouda P, Bellali S, Ngom II, et al. Extensive culturomics of 8 healthy samples enhances metagenomics efficiency. PLoS One. 2019;14:e0223543.

  • To use of italics for species names (see lines 262-270).

DONE as suggested.

  • To improve of the introduction by explaining why the authors focus the attention on bisphenol. In particular, I suggest to add a paragraph with references to policy regulations about (mis)use of bisphenols, as background information for the reader.

DONE as suggested, the following paragraphs explaining why the authors focus the attention on bisphenol and policy regulations related are summarized:

Bisphenols are considered as Microbiota Disrupting Chemicals (MDC) [5] and their presence in humans has been confirmed by detecting them in human biospecimens: feces, serum, urine, saliva, hair, tissue and blood [23,24]. Bisphenol A (BPA) is used in manufacturing polycarbonate and epoxy resins for food consumer products and packages. There is also cumulative exposure from contaminating soils, aquatic environments, drinking water, air and dust particles [25]. The estrogen activity alteration is the most widely studied effect of BPA and analogues, enhancing endocrine disruptor activities [26]. Moreover, some studies have shown obesogenic effects through microbiota dysbiosis [27], fat cell development and lipid accumulation [28]. There are several regulations enforced concerning the hazards of Bisphenol A, as derivative of polycarbonates plastics and epoxy resins, used in food contact materials, toys, or other products. In order to protect the consumers from cumulative exposure, the tolerable daily intake (TDI) for BPA is permanently re-evaluated according to new toxicity data through specific international projects, such as U.S. National Toxicology Program (CLARITY-BPA program) [29] or European Food Safety Authority (EFSA) comprehensive re-evaluation of BPA exposure and toxicity[30].

References included:

- Camacho L, Lewis SM, Vanlandingham MM, Olson GR, Davis KJ, Patton RE, Twaddle NC, Doerge DR, Churchwell MI, Bryant MS, McLellen FM, Woodling KA, Felton RP, Maisha MP, Juliar BE, Gamboa da Costa G, Delclos KB. A two-year toxicology study of bisphenol A (BPA) in Sprague-Dawley rats: CLARITY-BPA core study results. Food Chem Toxicol. 2019 Oct;132:110728. doi: 10.1016/j.fct.2019.110728. Epub 2019 Jul 28. PMID: 31365888.

- European Food Safety Authority (EFSA). Scientific Opinion on the risks to public health related to the presence of bisphenol A (BPA) in foodstuffs. EFSA Journal 2015;13(1):3978 (Assessed on 21 July, 2021; Available on:   https://doi.org/10.2903/j.efsa.2015.3978)

  • To add a conclusion paragraph/section, as the Abstract has a paragraph about "our conclusions" which is then not expanded in the manuscript. In this section, I recommend also to add considerations about what is missing to consider the described effects as "concrete", especially in the context of healthy both diet and life style.

Done as suggested, a section has been included

  1. Conclusions

Bacillus strains isolated from human gut microbiota, and taxonomically closest to the safe qualified B.subtilis and B. amyloliquefaciens groups, became a cultivable predominant taxa when high bisphenols exposure conditions were tested. In parallel, these strains harbored the PKS molecular gene biosynthetic loci and showed phenotypic antimicrobial effects. Therefore, they might be proposed as beneficial microorganisms with molecular features that would contribute to modulate the ecological taxa composition and funtionality of human gut microbiota. Intervention studies will be further needed to demonstrate the abilitity to recover microbiota dysbiosis triggered by high MDC exoposure diets and lifestyles towards eubiosis and healthier status.

  • To change of colours/patterns of bars represented in Figure 2, which I found difficult to be read. 

DONE as suggested.

Reviewer 2 Report

In this study, the authors enriched bacterial strains from human fecal cultures under the microbiota-disrupting chemical, bisphenol A (BPA). From the culture, 11 Bacillus strains were isolated from plates containing BPA and their antimicrobial activity was evaluated. Then, using whole genome sequence data, their function and safety as probiotics were confirmed. The authors proposed a novel method for selecting new bacterial species from human feces using a single model in this paper. This proposition is interesting, but the study is in a preliminary stage. I think more careful experimentation and detailed explanations are required.

  1. There are many pathogens in human feces, as the authors described for Bacillus cereus. The biggest problem for the development of probiotics is to clarify whether the bacterial strain is pathogenic or not. Even for the same bacteria, there are pathogenic and non-pathogenic strains, like Escherichia coli, and both of these strains are present in healthy human flora. Are the species and strains isolated in this study, Bacillus siamensis, B. velezensis, B. nematocidal, B. amyloliquefaciens and B. pacificus, safe bacteria?
  2. In this manuscript, there is no description of how the authors selected the 11 strains from the BPA-treated culture. The authors used different BPA concentrations (0.5, 10, 20 and 50 ppm). Were the bacterial strains concentrated in a medium in which the BPA concentration was gradually increased from 0.5 to 50 ppm? The authors described 27 % of the cultured bacteria consisted of Bacillus strains after BPA concentration. Why were Bacillus strains selected for probiotic bacteria?
  3. Enzymatic and antimicrobial activities of only 4 strains, B1, B3, B7 and B12, were determined. Why did the authors select the 4 strains from the 11 isolated strains? The most difficult step is how the strains are chosen from a panel consisting of many strains. I think new suggestions and discussion regarding this step constitute another subject of this study beside the enrichment of bacteria by BPA selection.
  4. There is no whole genome sequence and phenotypic analysis data for pacificus or B. nematocidal (B7 and B9.2 strain, respectively).
  5. Finally, the authors were searching for bacteria that degraded BPA (Fig. 2). However, they isolated BPA-tolerant strains. Do BPA-tolerant strains show high BPA-degrading activity?

Author Response

REVIEWERS COMMENTS:

AUTHORS ANSWERS

REVIEWER 2:

In this study, the authors enriched bacterial strains from human fecal cultures under the microbiota-disrupting chemical, bisphenol A (BPA). From the culture, 11 Bacillus strains were isolated from plates containing BPA and their antimicrobial activity was evaluated. Then, using whole genome sequence data, their function and safety as probiotics were confirmed. The authors proposed a novel method for selecting new bacterial species from human feces using a single model in this paper. This proposition is interesting, but the study is in a preliminary stage. I think more careful experimentation and detailed explanations are required.

Thank you very much for the overall evaluation of our manuscript.

  1. There are many pathogens in human feces, as the authors described for Bacillus cereus. The biggest problem for the development of probiotics is to clarify whether the bacterial strain is pathogenic or not. Even for the same bacteria, there are pathogenic and non-pathogenic strains, like Escherichia coli, and both of these strains are present in healthy human flora.

Are the species and strains isolated in this study, Bacillus siamensis, B. velezensis, B. nematocidal, B. amyloliquefaciens and B. pacificus, safe bacteria?

 The following paragraph has been included for addressing this important issue:

The strains isolated in the present work were clustered in the two main groups: B. subtilis–like (non-pathogenic) [59] and B. cereus-like (pathogenic) [60], as shown in Fig. 3, however the pathogenicity features are strain-specific dependent. The work approach is based on potential uses and predictive data analysis, but for further commercial uses, a safety assessment should be done for each strain, to demonstrate that do not pose any safety and/or pathogenicity concerns. The battery of tests usually requested is: antibiotic resistance test no greater than existing regulatory cutoffs against clinically important antibiotics, not capacity of inducing hemolysis or producing surfactant factors, and the absence of virulence or toxigenic activity in vitro.

-Bianco A, Capozzi L, Monno MR, Del Sambro L, Manzulli V, Pesole G, Loconsole D, Parisi A. Characterization of Bacillus cereus Group Isolates From Human Bacteremia by Whole-Genome Sequencing. Front Microbiol. 2021 Jan 12;11:599524. doi: 10.3389/fmicb.2020.599524. PMID: 33510722; PMCID: PMC7835510.

-Harwood CR, Mouillon JM, Pohl S, Arnau J. Secondary metabolite production and the safety of industrially important members of the Bacillus subtilis group. FEMS Microbiol Rev. 2018 Nov 1;42(6):721-738. doi: 10.1093/femsre/fuy028. PMID: 30053041; PMCID: PMC6199538.

 2. In this manuscript, there is no description of how the authors selected the 11 strains from the BPA-treated culture. The authors used different BPA concentrations (0.5, 10, 20 and 50 ppm). 

  • Were the bacterial strains concentrated in a medium in which the BPA concentration was gradually increased from 0.5 to 50 ppm?

 A new phrase addresses this issue; please see the new readable paragraph:

Ten isolates from fecal human microbiota collections of 0 to 1 year old infants (Isolates B-Project INFABIO) appropriately maintained at -80 ºC underwent a directed culturing approach using 0.5 g of the fecal specimen in 1.5 mL of Brain Heart Infusion or Man Rogosa and Sharpe (BHI/MRS) broths, adding different concentrations of BPA [0.5, 10, 20, and 50 ppm], in order to search tolerant and/or potentially BPA biodegrading microorganisms, incubation for 72 h. Further serial dilutions and spreading onto BHI/MRS solid media plus incubation under aerobic and anaerobic conditions (anaerobic jars anaerocult®) at 37 ºC during 72 h were applied. BPA-tolerant colonies with distinguishing features were isolated as pure culture for subsequent morphological, phenotypic and genotypic identifications: bacterial cell counts, gram staining, spore staining, capsule staining, catalase activity, oxidase, and motility tests.

  • The authors described 27 % of the cultured bacteria consisted of Bacillusstrains after BPA concentration. Why were Bacillus strains selected for probiotic bacteria?

 Explanation: During last decade a triplication of studies related to Bacillus strains proposed for being used as human, animal or plant probiotics are available. Please see the scheme showing the exponential growing trend of scientific evidence for “Bacillus and Probiotics” in Pubmed.  Fig. attached

We have added a paragraph to draw the attention to this fact:

Recently, several Bacillus strains have been extensively proposed for being used as human and animal probiotics [55-56]. Most of the species used are belonging to Bacillus subtilis and Bacillus amyloliquefaciens groups and special attention deserved the food and clinical studies with strains that showed special enzyme capacities [57] or are able to modulate and mitigating pathophysiological disorders  [58].

- Szlufman C, Shemesh M. Role of Probiotic Bacilli in Developing Synbiotic Food: Challenges and Opportunities. Front Microbiol. 2021 Apr 12;12:638830. doi: 10.3389/fmicb.2021.638830. eCollection 2021. PMID: 33912147

- Khalid F, Khalid A, Fu Y, Hu Q, Zheng Y, Khan S, Wang Z. Potential of Bacillus velezensis as a probiotic in animal feed: a review. J Microbiol. 2021 Jul;59(7):627-633. doi: 10.1007/s12275-021-1161-1. Epub 2021 Jul 1. PMID: 34212287 Review.

-Sultana OF, Lee S, Seo H, Mahmud HA, Kim S, Seo A, Kim M, Song HY. Biodegradation and removal of PAH by Bacillus velezensis isolated from fermented food. J Microbiol Biotechnol. 2021 May 21;31(6). doi: 10.4014/jmb.2104.04023. Online ahead of print. PMID: 34024889

- Kang M, Choi HJ, Yun B, Lee J, Yoo J, Yang HJ, Jeong DY, Kim Y, Oh S. Bacillus amyloliquefaciens SCGB1 Alleviates Dextran Sulfate Sodium-Induced Colitis in Mice Through Immune Regulation. J Med Food. 2021 Jul;24(7):709-719. doi: 10.1089/jmf.2021.K.0044. PMID: 34280033

3. Enzymatic and antimicrobial activities of only 4 strains, B1, B3, B7 and B12, were determined. 

  • Why did the authors select the 4 strains from the 11 isolated strains? The most difficult step is how the strains are chosen from a panel consisting of many strains. I think new suggestions and discussion regarding this step constitute another subject of this study beside the enrichment of bacteria by BPA selection.

 Explanation-WGS analysis: Preliminary antibiotic production results grouped the strains in four groups (B1+B2; B2+B3; B7; B12) with very similar results that were in agreement with main taxonomic clusters.

 Therefore, in order to do more comprehensive manuscript and tables, we only show relevant antibiotic production results in Table 4.  

 Text inserted: Preliminary results grouped the strains according to their capacity of antibiotic production with very similar inhibiting zone value, which were also in agreement with the main taxonomic clusters.They were organized as follow: rB1 represented B1, B4, B5, B6, B7, B8, B9, and B9.2; rB3 represented B2 and B3; rB7 represented B7 and B12.

4. There is no whole genome sequence and phenotypic analysis data for B.pacificus or B. nematocidal (B7 and B9.2 strain, respectively).

Explanation-WGS analysis:  It was done the Whole genome sequence analysis for B. pacificus and B. nematocida as stated under the table 6. Moreover, we have highlighted and include a reference:

 Please see the highlighted notes under the Table 6.

B.pacificus (strain R1) NC_NJQG01000001 Loci was not found

B.nematocida (strain B-16T) No WGS is available – Analysis PKS Loci was not applicable

Reference:  Huang XW, Niu QH, Zhou W, Zhang KQ. Bacillus nematocida sp. nov., a novel bacterial strain with nematotoxic activity isolated from soil in Yunnan, China. Syst Appl Microbiol. 2005 Jun;28(4):323-7. doi: 10.1016/j.syapm.2005.01.008. PMID: 15997705.

Explanation-Phenotypic analyses:

B7 was performed and data are shown in Table 4. No inhibition zone/halo were produced as it is indicated (-)

B9.2 was preliminary performed and equivalent data to the representative taxonomical strain B1 were found.

5. Finally, the authors were searching for bacteria that degraded BPA (Fig. 2). However, they isolated BPA-tolerant strains. Do BPA-tolerant strains show high BPA-degrading activity?

Explanation: All the strains isolated and described were experimentally tolerating different concentrations of BPA and growing efficiently. However, BPA biodegradation can be only demonstrated through identifying specifically the genes encoding enzymes involved on BPA degradation pathways (data available for theoretical predictive analysis not shown because we considered that were beyond the aim of this study).

Round 2

Reviewer 2 Report

Significant figures were not considered for the values listed in Table 4. This point should be revised before accepting the manuscript.

Author Response

REVIEWERS COMMENTS:

AUTHORS ANSWERS

REVIEWER 2:

Significant figures were not considered for the values listed in Table 4. This point should be revised before accepting the manuscript.

Thank you very much for this relevant observation. Please see the ammendements performed to improve the manuscript according to your suggestions:

Material and Methods

The data were expressed as mean of the three replicates. Tests were done spreading the indicator microbial strains over the surface of the Müller-Hinton agar using sterile cotton swab. Inside six mm diameter oxford wells generated in agar, 20 µl of antibiotic producing bacteria extract was added. Standards appropriate positive controls (ampicillin, gentamycin and streptomycin at 10 µg) and negative/blank (sterile media/ethanol) were used. The plates were incubated at 37 °C for 24 h and the inhibition zones were measured.

Table 4. Antimicrobial activity of BPA-tolerant human gut microbiota isolated strains.

Target Indicator Bacteria

Strains rB1

Strains rB3

Strains rB7

Diameter of inhibitory zone (mm)±SD1

Bacillus cereus

15 ± 0

17 ± 0

-          

Bacillus circulans

13 ± 0

14,3 ± 1,2

-

Staphylococcus aureus

11,7 ± 0,6

10 ± 0

-

Streptococcus pyogenes

15 ± 0

13,3 ± 0,6

-

Serratia marcescens

17 ± 0

15,3 ± 1,5

-

E. coli

15 ± 0

13,3 ± 0,6

-

Salmonella

11 ± 0

10 ± 0

-

Klebsiella

20 ± 0*

15 ± 0*

-

Pseudomonas

-

-

-

1Values are mean diameter of inhibitory zone (mm)±SD of three replicates. The diameter of well (6 mm) was included.

(–) Diameter of inhibitory zone <7 mm considered as no antimicrobial activity.

*Significant values compared to theroretical values from B. subtilis polyketides [64]

Minimum inhibitory concentration (MIC) values were similar to those resultant of other polyketides antimicrobial effects previously described, being significant differential and higher the effects found against Klebsiella [64]. Therefore, searching putative biosynthetic pathway of the pks gene product proceeded after the validated molecular antimicrobial attributions.